# Subject Separation Network for Reducing Calibration Time of MI-Based BCI

**DOI:** 10.3390/brainsci13020221

**Published:** 2023-01-28

**Authors:** Haochen Hu, Kang Yue, Mei Guo, Kai Lu, Yue Liu

**Affiliations:** 1Beijing Engineering Research Center of Mixed Reality and Advanced Display, School of Optics and Photonics, Beijing Institute of Technology, Beijing 100081, China; 2Institute of Software, Chinese Academy of Sciences, Beijing 100045, China

**Keywords:** Brain Computer Interface, deep learning, motor imagery, domain adaptation, calibration reduction

## Abstract

Motor imagery brain–computer interface (MI-based BCIs) have demonstrated great potential in various applications. However, to well generalize classifiers to new subjects, a time-consuming calibration process is necessary due to high inter-subject variabilities of EEG signals. This process is costly and tedious, hindering the further expansion of MI-based BCIs outside of the laboratory. To reduce the calibration time of MI-based BCIs, we propose a novel domain adaptation framework that adapts multiple source subjects’ labeled data to the unseen trials of target subjects. Firstly, we train one Subject Separation Network(SSN) for each of the source subjects in the dataset. Based on adversarial domain adaptation, a shared encoder is constructed to learn similar representations for both domains. Secondly, to model the factors that cause subject variabilities and eliminate the correlated noise existing in common feature space, private feature spaces orthogonal to the shared counterpart are learned for each subject. We use a shared decoder to validate that the model is actually learning from task-relevant neurophysiological information. At last, an ensemble classifier is built by the integration of the SSNs using the information extracted from each subject’s task-relevant characteristics. To quantify the efficacy of the framework, we analyze the accuracy–calibration cost trade-off in MI-based BCIs, and theoretically guarantee a generalization bound on the target error. Visualizations of the transformed features illustrate the effectiveness of domain adaptation. The experimental results on the BCI Competition IV-IIa dataset prove the effectiveness of the proposed framework compared with multiple classification methods. We infer from our results that users could learn to control MI-based BCIs without a heavy calibration process. Our study further shows how to design and train Neural Networks to decode task-related information from different subjects and highlights the potential of deep learning methods for inter-subject EEG decoding.

## 1. Introduction

Brain Computer Interfaces (BCIs) provide a direct path to link a user’s brain and an external device, surpassing peripheral nerves and muscles [1]. BCI based on Motor Imagery (MI) is one of the most widely researched paradigms available to transform user motor intentions into control signals [2]. Subjects can transmit their intention to an external device, such as an exoskeleton robot [3] or wheelchair, by simply imagining the movement of different body parts, which can help neuromuscular injury patients to restore their motor abilities or facilitate their daily life [4]. Multiple techniques have been developed to record subject’s brain activities. Among them, electroencephalogram (EEG) has received wide attention due to its non-invasive nature and high temporal resolution, and has been employed in various BCI applications. BCIs can also be used as an assistive technology to communicate and interact with the surrounding environment without the help of an intermediator (e.g., a caregiver or a nurse), which poses significant clinical implications for motor-impaired patients. Ref. [5] proposed a Neural Internet as a system which transforms brain signals into computer commands so as to help locked-in patients to surf the internet. They introduced a prototype of EEG-controlled web browser operated by self SCP regulation, achieving a mean classification accuracy of 80%. Ref. [6] further extended the EEG-controlled browser by introducing email and virtual keyboard to widen the range of hypertext-based applications available, as well as by using graphical in-place markers to let users select any link on a web page.

Despite the achievements of MI-based BCIs, they still suffer from the problem of long calibration times and suboptimal decoding accuracy. Traditional EEG decoding methods mainly consist of several stages, including signal preprocessing, feature extraction and classification [7]. It is a common practice to collect a few runs of a new subject’s motor imagery data for training a subject-specific classifier before the BCI system can produce accurate commands [8]. This process, often referred to as the calibration phase of a BCI, is time-consuming and tedious due to the complex configuration of EEG headsets and long periods of modulating neural activities that cause fatigue [9].

To reduce the calibration times of MI-based BCIs, several domain adaptation methods have been introduced to facilitate the utilization of pre-recorded EEG data obtained from other subjects [10]. The objective of domain adaptation is to extract knowledge from a source domain to accomplish tasks in the target domain with a limited amount of data [11], which is highly in accordance with the situation of training classifiers for MI-based BCIs. By gathering labeled trials in a source domain to train models that can well generalize to target subjects’ data, we can greatly reduce the amount of data needed for the calibration process. This approach is based on the fact that most subjects’ EEG signals share the same phenomenon of event-related desynchronization (ERD) when they are imagining certain kind of motor behavior [12]. However, the performance of this transfer process will deteriorate if the target subject’s EEG pattern significantly lacks discriminative information. Traditional machine learning models lack the capabilities of representing a hypothesis space large enough to fit those outliers of high-dimensional data.

On the other hand, deep learning model simplify the complex feature engineering process of traditional machine learning, demonstrating strong learning ability for ultra-high-dimensional data [13]. Different forms of neural network are widely used in the classification of image, video, audio and other signals. Deep learning methods can be used to handle domain adaptation tasks, and reduce the risk of neglecting certain subjects’ distributions. The reason is that a neural network with great depth is capable of restoring a huge amount of information [14]. Nonetheless, the methods based on deep learning applied to EEG signals also face several challenges [15]. Specifically, the powerful fitting ability and insufficiency of training data make neural networks prone to overfitting, and the features learned by using a deep learning model face the problem of lacking interpretability compared with hand-crafted features of traditional machine learning methods [16].

To address the above problems, we proposed an ensemble Subject Separation Network (eSSN) domain adaptation framework to reduce the necessity to collect calibration EEG data for new subjects of MI-based BCI. This framework consists of two stages. Firstly, a set of networks, called SSNs, is trained on each source subject’s labeled data and a small amount of the target subject’s unlabeled data. For each individual SSN, domain adversarial training is used to transform the target subject’s features to the source domain. Secondly, after training for each distinct source–target subject pair network, all SSNs are aggregated by using an ensemble classifier for a final prediction.

The main contributions of this paper are listed as follows:We propose a neural network based on domain adaptation, namely the ensemble Subject Separation Network, to reduce calibration time by leveraging other subjects’ data.We propose the definition of Session-ITR as the theoretical basis and interpretation tool for the performance of cross-subject EEG decoding algorithms.We evaluate multiple network design choices, and compare our method with several state-of-the-art EEG decoding methods. Evaluation results prove that the eSSN significantly outperforms other methods in terms of accuracy and Session-ITR.

## 2. Related Work

### 2.1. Neurophysiological Background

Motion Imagery can generate detectable patterns in both primary sensor area and primary motor area. These patterns are related to the motor preparation process in which mu (8–12 Hz) and central beta rhythms (13–28 Hz) go through desynchronization similar to a real motor execution process [17]. The BCI system based on motor imagery (MI-BCI) is usually constructed according to this phenomenon.

However, for each new subject participating in the experiment, the EEG data of the subject must be collected to train a new classifier, which usually entails a tedious process. The reason for this is that inter-subject variation exists in the EEG patterns produced by subjects’ motor imagery processes [18]. The factors causing inter-subject variation of MI-BCIs range from neurophysiological structure to the experimental paradigm. They can also be categorized into neuroanatomical factors, psychological factors and neurophysiological factors.

Neuroanatomical factors are static and determined by the subject’s brain structure, which can be measured by using a variety of real-time brain imaging methods. Ref. [19] performed whole-brain voxel-based morphometry (VBM) on a group of BCI subjects using mu wave-based control cursors, and found that the activities of left dorsal premotor cortex (pre-PMd), right supplementary motor area (SMA) and right supplementary somatosensory area (SSA) have a positive relationship with the performance of the BCI. Similarly, Ref. [20] used MRI to prove that the heterosexual scores of cingulum, superior front occipital fascicle and corpus callosum were related to the performance of the BCI. In addition, in Ref. [21], it was found that the number of voxels activated by frontal gyrus responsible for motor observation was correlated with BCI performance (r = 0.72). These neuroanatomical variables are usually difficult to observe.

Psychological factors include the subject’s motor imagery strategy and the degree of participation. These factors can be divided into relatively static user capabilities and relatively dynamic experimental-related factors. The former includes reaction-time and motor imagery strategy. It is proposed in [22] that the response time (RT) of subjects to specific stimuli can be used to predict the BCI performance of subjects. Under four different feedback update frequencies (FUI), RT was found to be negatively correlated with BCI performance. Ref. [23] studied the effects of two different motor imagery strategies (kinesthetic and visual-motor) on the performance of MI-BCIs and found that only the former can form stable and resolvable EEG patterns. On the other hand, experiment settings and user’s arousal state will affect the final performance. In Ref. [24], it is proposed that neural feedback is available to adjust the arousal level of subjects, so as to optimize the operation performance.

Neurophysiological factors include EEG-based features, e.g., mu wave amplitude, which are also called performance predictors. The significance of predictors is multifaceted, that is, subjects can be pre-screened according to these features, and appropriate classifiers can be selected for the target subjects according to these features. Comparatively, these factors are easy to observe, but they need deeper theoretical support for their interpretation.

One of the predictors is the mu wave predictor. Mu waves are brain waves at the frequency of 8–15 Hz in the sensorimotor region, which is insensitive to visual input. In fact, mu waves contain many components with different topologies and functions. In the postcentral somatosensory cortex, the representation areas of different body parts may produce different patterns and superimpose them, in which the proportion of the hand is the largest [25]. Another related brain wave is the beta band (16–30 Hz) of a resting EEG, which exists in the central motor cortex.

Previous work shows that the higher the mu wave band power of a resting EEG, the better the BCI performance. In [26], Bayesian spatio-spectral filter optimization (BSSFO) was proposed to obtain the frequency band characteristics of subjects, and cluster them according to the characteristics of subjects. The 2-minute 3-channel resting EEG data of each subject were used to predict the BCI performance of the subjects, which shows that the characteristics of subject’s mu wave activities contain the information regarding the subject’s ability to operate an MI-based BCI system. We will use this predictor to visualize the inter-subject variation of Motor Imagery EEG signals in Section 4.4.

### 2.2. Domain Adaptation in EEG Decoding

The inter-subject variation of MI-based BCIs is more significant than those of reactive BCIs, such as SSVEP BCIs. Therefore, various methods have been developed for cross-subject BCI classification algorithms [27,28]. Domain adaptation is dedicated to solving knowledge transfer in different domain distributions. As a transfer learning method, domain adaptation can be further divided into feature modeling and instance modeling [29]. The implementation methods include adversarial learning and reconstruction of source signals.

Instance-based methods are developed according to the insight that there is a design space for the selection of the training set. Algorithm [30] uses MTS (manifold trial selection) to select samples which are useful for transfer learning, and classifies MI data on the tangent space of a Symmetric Positive Definite Manifold. Algorithm [31] selects the source subjects closest to the target subjects through the spatial-temporal characteristics of the resting EEG, and uses deep adversarial learning to narrow the difference between the characteristics of the source domain and target domain.

A feature-based domain adaptation method attempts to map the features of the target domain and the source domain to the same space. For this purpose, we not only need to design the metric to measure the distance between features and add it to the objective function as a regularization term, but also select the optimization algorithm and function family to reduce this distance. Algorithm [32] proposed such a standard joint distribution density (JDD), which can simultaneously align the edge probability distribution and conditional probability distribution of two domains. Then, feature learning and classification are carried out based on deep learning. Algorithm [33] proposed a new countermeasure method and conditional domain adaptation network (cDAN) to restrict feature differences. Similarly, Algorithm [34] and Algorithm [28] also use the confrontational training method.

The proposed method of eSSN utilized the aforementioned adversarial learning method for feature modeling. To be specific, a shared feature space of both source and target domain is constructed by adversarially training a Shared Encoder—Domain Classifier pair. Feature modeling can also be achieved by adding a specific loss to the feature space. The implementation of JDD loss mentioned above is inappropriate since we have only unlabeled target data. Therefore, we re-implement the widely used Minimum Mean Discrepancy (MMD) loss instead in an ablation study. The mechanism of instance transfer is not adopted, since further selection of the relatively small training set may lead to overfitting of deep networks.

## 3. Materials and Methods

Inspired by [35], we propose a Subject Separation Network that explicitly models the unique characteristics of subject’s ERD patterns while mapping their EEG trials to the same feature space. Given a source dataset containing labeled data from multiple subjects and a target dataset of new subject’s unlabeled trials, the task of the Subject Separation Network is to train an end-to-end EEG classifier on a source dataset that can be generalized well to the target dataset.

### 3.1. Problem Definition

Let $D_{S}={\left\{X_{s}\right\}}_{s=0}^{N_{s}}$ represent the source dataset and $X_{s}={\left\{x_{i}^{s},y_{i}^{s}\right\}}_{i=0}^{N_{s}}$ represent each sub dataset containing one subject’s labeled data. Let $X_{T}={\left\{x_{i}\right\}}_{i=0}^{N_{t}}$ represent an unlabeled dataset of the target subject. The objective of SSN is to learn a classifier trained on labeled trials from subset $X_{s}$ and unlabeled data from target dataset $X_{T}$. After a total of $N_{s}$ SSNs corresponding to $N_{s}$ source subjects are built, they are assembled to form a final classifier, namely the eSSN. The ensemble classifier’s performance is then evaluated on $X_{T}$ by comparing with baseline methods. The terms of domain and subject are used interchangeably in this paper, as each subject is regarded as a separate domain.

### 3.2. Datasets

The dataset consists of nine healthy subjects’ motor imagery data, including 22 channels of EEG signal and 3 channels of electro-oculogram signals at 250 Hz. There are four classes of motor imagery tasks, namely imagination of left hand movement, imagination of right hand movement, imagination of tongue movement and imagination of feet movement. Each class contains 72 trials divided into two sessions. We intercept the data from 0.5 to 4.5 s after the display of motor imagery cue as the input EEG data. Further, band-pass filtering of 4–38 Hz is performed on the raw EEG data to remove DC offsets and power-line noise. An exponential moving standardized procedure is conducted as described in the works of [36]. Finally, the data is downsampled from 250 to 50 Hz to lower the computation cost and GPU storage.

### 3.3. Model Architecture

As shown in Figure 1, the source domain data and target domain data will separately enter the eSSN network. As mentioned in the previous section, an SSN is constructed for each source domain using both labeled source data and unlabeled target information. The shared encoder learns from both domain subject-invariant task-related common features, which are used by the classifier for the prediction of EEG class. Two separate private feature spaces are learned from each domain to avoid task-unrelated noise. A shared decoder is concatenated to the private and shared encoder to ensure that both feature extractors learn from meaningful physiological information involved in the EEG signal. After all losses are converged for each SSN, all SSNs then vote equally to form a final decision.

#### 3.3.1. Shared Encoder

The shared encoder is developed so that the features from source and target domain should share similar representations. The word “shared” means that this part of the network will be shared by source and target domain, while the network components with the prefix “private” have different copies for each domain and produce isolated features.

We take the separate spatial-temporal convolution architecture as the feature extractor for application. As shown in Figure 2, a 2D convolution is first applied to input signal *x* with *c* channels and *t* time steps. The convolution is two dimensional; therefore, all channels share the same group of 1D convolution parameters during each operation. This operation can be regarded as passing the preprocessed EEG signal through a set of time filters whose spectral characteristics are controlled by the weights of the convolution layer. Lower level spectral information is mainly extracted by this layer and stored in the group of features. After that, a 2D convolution layer is concatenated to extract spatial-related information from the preprocessed signal. A filter with the kernel size of *c* goes through feature maps of the previous layer to produce a set of spatial abstraction of each spectral group. The spectrum of different classes of motor imagery signal are known to have discriminative spatial distributions. This phenomenon is utilized by most traditional EEG signal classification methods with either hand-crafted or data-driven spatial filters that assign distinct values for different parts of the cortex. The combined spatial-spectral feature then passes through a batch normalization layer and a non-linear layer, which is common in convolution network designs. A pooling layer is used to produce a summation of the combined feature. The reason for the adoption of this layer is two-fold. The reduced dimension of final representation helps the network to minimize the possibility of overfitting, and the nature of the pooling layer makes the discriminative information in the combined feature time-invariant. At last, another non-linear layer and a dropout layer are placed in front of the end of the extractor.

#### 3.3.2. Domain Classifier

The shared encoder is required to learn similar representations for source and target domain’s EEG trials. There are several methods to achieve this goal in domain adaptation; among them, we experiment with two common methods, i.e., domain adversarial learning and MMD constraint. We adopt adversarial learning to transform the source feature distribution to the target domain.

The adversarial learning method operates mainly on two parts, that is, the domain classifier that manages to predict the input feature domain, and a feature extractor that tries to fool the classifier by producing the former domain-invariant features. These domain-invariant feature will significantly improve the decoding accuracy of the target domain’s data.

We choose the shared encoder whose output is used to discriminate between different motor imagery classes as the feature extractor of the adversarial learning framework. As for the domain classifier, we apply a simple architecture of a fully-connected layer whose output is mapped to the class of different domains. In our paper, the domain class only has two values, representing either source or target domain.

#### 3.3.3. Private Encoder

According to [35], the features trivially learned under similarity constraint may include the noise highly correlated with the shared representation. Inspired by [35], a private encoder is introduced to produce a separate representation for each domain. For each domain, we maintain a subspace that is orthogonal to the shared feature space constraint by an orthogonal loss function. By applying such a constraint, the classifier is able to learn from the shared feature uncontaminated by each domain’s specific pattern.

The private and shared encoder share the same network architecture. The reason is that this network backbone is capable of extracting task-related motor imagery features while avoiding overfitting by applying a separate spatial-temporal convolution.

#### 3.3.4. Shared Decoder

To ensure that both the shared and private representations are built on meaningful characteristics extracted from the EEG signal, the combination of shared features and private features should be informative enough to reconstruct the original signal. Therefore, a shared decoder is built to generate a synthetic signal based on the combination of private and shared feature. Further, a reconstruction loss is added to avoid trivial solutions by making the reconstructed signal more similar to the input signal.

For the shared decoder, we first apply a deconvolution layer to the learned feature to restore the channel information, the reason is that the spatial convolution comes after temporal convolution during feature extraction. Then, a trivial linear transformation is applied to each channel feature to reconstruct the EEG signal. As shown in Figure 3, the shared decoder takes both the private feature and shared feature as input, and the output has the same shape as the SSN network input.

#### 3.3.5. Ensemble Learning

When obtaining each source–target subject pair SSN network, as explained in previous sections, all $N_{s}$ SSNs are gathered to construct an ensemble classifier for the final classification decision. We apply a trial method of voting classifier, that is, each SSN gives an equal vote to its predicted class, and the class that wins the most votes is taken as the final output.

### 3.4. Loss Function

The objective of the network is to build a function $F_{c}\left(x;\theta_{c}\right)$ with learnable parameter $\theta_{c}$ that maps an EEG trial *x* to a shared feature space $f_{c}$, under the constraint that an analogous function $F_{p}\left(x;\theta_{p}\right)$ orthogonally transforms *x* to a private feature space $f_{p}$, and a function $R(f;\theta_{r})$ reconstructs the EEG signal from the combined features of $f_{c}$ and $f_{p}$. To ensure that source and target domain share similar representations, a domain classifier $D(f;\theta_{d})$ is introduced. By trying to predict the feature’s domain, the domain classifier acts as the shared encoder $F_{c}\left(x;\theta_{c}\right)$ that tries to find similar representations for both domains.

The objective of the training process is to optimize the following loss:(1)$$L=L_{task}+\alpha_{r}L_{recons}+\alpha_{d}L_{difference}+\alpha_{s}L_{similarity}$$

The task in the first loss item refers to the objective of the classifier trained on the source subject’s data to well generalize to the new subject’s EEG trials. The parameters $\alpha_{r},\alpha_{d},\alpha_{s}$ are hyper parameters that need to be adjusted. Therefore, we minimize the empirical negative log likelihood of the ground truth class for each source domain where we have label information:(2)$$L_{task}=-\Sigma_{i=0}^{N_{s}}y_{i}^{s}log\left({\hat{y}}_{i}^{s}\right)$$
where $y_{i}^{s}$ refers to the one-shot encoding of class label from source domain *s* and input *i*. ${\hat{y}}_{i}^{s}$ represents the SoftMax predictions of the model. For construction loss, we apply mean squared error to optimize the reconstruction process:(3)$$L_{recon}=\frac{1}{Ns}\Sigma_{i=0}^{N_{s}}||x-\hat{x}{\left|\right|}_{2}^{2}+\frac{1}{Nt}\Sigma_{i=0}^{N_{t}}||x-\hat{x}{\left|\right|}_{2}^{2}$$
where $\hat{x}$ is the mean of all reconstructed trials. To separate each subject’s own feature space, a difference loss is added to the outputs of the shared and private encoder, as shown in Figure 3. This loss encourages the network to learn orthogonal solutions for each feature space:(4)$$L_{difference}=\left|\right|H_{c}^{sT}H_{p}^{s}{\left|\right|}_{F}^{2}+\left|\right|H_{c}^{tT}H_{p}^{t}{\left|\right|}_{F}^{2}$$
where $H_{p}^{s}$ refers to private features from the source subject and $H_{c}^{t}$ refers to the shared features from the target subject.

### 3.5. Theoretical Consideration

The main purpose of domain adaptation is to optimize the generalization ability of the network for the target domain with a minimum amount of the target domain’s labeled data, while the network is mainly trained on source domain data. In theory, label information in the target domain is not required in the training process. At the completion of the establishment of the eSSN model, the remaining target subject’s labeled trials are used as a test dataset to evaluate the performance of the model, as demonstrated in Section 3.1. The model with higher performance may need less calibration data for fine-tuning, thereby directly reducing the calibration time.

We take the analysis to the next step by introducing a measure called Mean Session *ITR* (MS-*ITR*). Though accuracy and other existing metrics (e.g., kappa value) can effectively represent the classifier’s ability to discriminate single-trial EEG signals, they are insufficient to represent the classifier’s ability to generalize over different subjects. The reason is that the cost of information required to obtain such an accuracy is not considered. To be more specific, let ITR(n) represent the BCI system’s mean ITR when using *n* labeled trials as training data. Higher ITR(n) with minimal *n* represents higher performance of the BCI system with a minimal amount of calibration data. $ITR_{train}$ stands for the information output during the training process, and the classifier has accuracy at chance-level. Since the Information Transfer Rate (ITR) of a BCI system [37] is defined as:(5)$$ITR_{trial}\left(\epsilon \right)=log_{2}M+\left(1-\epsilon \right)log_{2}\left(1-\epsilon \right)+\epsilon log_{2}\frac{\epsilon }{M-1}$$
where *M* refers to the number of class. As $\epsilon =\frac{M-1}{M}$ during training, we then obtain $ITR_{train}=0$. The ITR for any portion of the training data could have contributed to ITR(n) if they were not chosen as training data. As a consequence, the amount of ITR “wasted” on the training phase is proportional to *n*. On the other hand, larger *n* indicates a longer period of calibration, which may lead to higher ITR in subsequent sessions. To evaluate the classifier’s generalization ability considering the above dilemma, the concept of Mean Session *ITR* (MS-*ITR*) is proposed to measure the model’s accuracy and the amount of calibration data required by that accuracy at the same time.

Mean Session *ITR* (MS-*ITR*) is defined to measure the performance of a cross-subject BCI algorithm. Suppose the first *n* trials are used for calibration, the remaining N−n trials have an accuracy of 1−ϵ, then Mean Session *ITR* can be estimated as:(6)$$ITR_{MS}\left(n,\epsilon \right)=\frac{1}{N}ITR_{trial}\left(\epsilon \right)\times \left(N-n\right)=\frac{N-n}{N}\left(log_{2}M+\left(1-\epsilon \right)log_{2}\left(1-\epsilon \right)+\epsilon log_{2}\frac{\epsilon }{M-1}\right)$$

As mentioned before, the accuracy of the trained model will also increase with the increase of *n*. In theory, there is an upper bound due to the differences that exist between subjects and machine learning algorithms. According to [38], supposing that the VC dimension of model’s hypothesis space *h* is *d*, $U_{T},U_{S}$ is the unlabeled data sampled from $D_{T},D_{S}$ (m’ in total), and *S* is the labeled data sampled from $D_{T},D_{S}$ (a total of βn and (1−β)n), for any δ∈(0,1), the following formula holds with the minimum probability of 1−δ:(7)$$\epsilon_{T}\left(\hat{h}\right)\leq \epsilon_{T}\left(h_{T}^{*}\right)+4\sqrt{\frac{\alpha^{2}}{\beta }+\frac{{\left(1-\alpha \right)}^{2}}{1-\beta }}\sqrt{\frac{2d\times log\left(\left(2n+1\right)\right)+log\left(\frac{2}{\delta }\right)}{n}}+C_{0}$$

The upper limit of the error rate decreases with the increase of *N*, which is a monotonic function of *n*. On the other hand, because $ITR_{trial}$ is a monotonic function of ϵ, the upper bound of Mean Session *ITR* can be determined as:(8)$$ITR_{MS}\leq \frac{N-n}{N}\left(log_{2}M+\left(1-f\left(\alpha ,n\right)\right)log_{2}\left(1-f\left(\alpha ,n\right)\right)+f\left(\alpha ,n\right)log_{2}\frac{f(\alpha ,n)}{M-1}\right)$$

Intuitively, considering the relationship between the error rate ϵ and the data *n* participating in the calibration, our optimization idea is to use the minimum amount of data *n* to make ϵ reach the lower limit.

## 4. Results

To verify the effectiveness of the proposed method, a cross-subject EEG decoding experiment is conducted on Dataset 2a from BCI Competition IV [39]. The experiment consists of nine rounds of training and evaluation. In each round, one subject is chosen as the target subject and the rest are taken as the source subjects. Classifiers are built based on the labeled data from the source subjects and the unlabeled data from the target subject, which are finally evaluated on the target subject’s labeled data. We first split all labeled data from the target subject into a training set and test set where 70% of the data are chosen as training data. Then, we remove the label information from the training set and add them to the domain adaptation process. The test set, which is never touched by the trained model, will be used to evaluate the model’s cross-subject accuracy. We also conduct a within-subject EEG decoding experiment to testify to the effectiveness of baseline methods of classifying EEG signals in a subject-specific scenario.

### 4.1. Baseline Algorithms

Our proposed eSSN method is compared with the following algorithms:Filter-Bank Common Spatial Pattern (FBCSP) [40], winner of several BCI Competitions [39], which is used as the baseline algorithm for decoding motor imagery EEG signals. It can automatically find each filter bank’s CSP filter features and discriminative subset to reduce the dimension of final features and prevent overfitting. Further, Support Vector Machine (SVM) is utilized for classification. It is worth noting that the EEG data does not go through the band-pass filter preprocessing as mentioned above, because this process is embedded in FBCSP itself.CSSP [41] is one of the variants of the classic CSP algorithm. By utilizing a spatial filter on a delayed signal as parameterized temporal filters, CSSP can extract joint spatial-spectral discriminative features. In addition, Linear Discriminative Analysis (LDA) is used for classification.Shallow ConvNet is proposed in [36]. In this work, multiple network architectures and a large space of hyperparameters are searched for the construction of a robust EEG decoding network. Shallow ConvNet enjoys outstanding performance in most comparison experiments; therefore, it is chosen as baseline algorithm.EEGNet is proposed in [42], and uses a stacked convolution network as the feature extractor similar to Shallow ConvNet, but adopting a better variant of depth-wise convolution in its structure. This architecture offers EEGNet the ability to achieve higher representative power with fewer trainable parameters, thereby effectively preventing overfitting.CDAN is an EEG classifier adversarially based on domain. A conditional domain discriminator is applied to learn high-level discriminative features [43]. This method has achieved competitive experimental results on the High Gamma Dataset [36], which is chosen as our baseline method.

Since we lack the access to the source code of FBCSP, CSSP and CDAN methods, these models are re-implemented in Python, which is a language widely used by deep-learning researchers. All the deep-learning models are implemented by using Pytorch, while others use Scikit-learn. The EEG data preprocessing pipelines are mostly implemented by using NumPy and the MNE library.

### 4.2. Classification Results

It can be observed from Figure 4a that the proposed eSSN method achieves the best cross-subject accuracy of 57.65% among the competing algorithms. For subject 1 and subject 9, eSSN achieves accuracy of 75.86% and 67.24% on target subjects’ labeled trials. As shown in Figure 4a, this level of performance is comparable if not better than the performance of several models in subject-specific scenarios. This result indicates the capability of the proposed method to achieve high-level performance on target domain data by integrating information from other subject domains through domain adaptation.

As shown in Figure 4a, the baseline methods EEGNet and ShallowConvNet achieve mean accuracy of 26.63% and 28.35%, respectively. While traditional machine learning methods FBCSP and CSSP gain suboptimal mean accuracies of 36.01% and 28.54%. To further confirm the effectiveness of baseline methods in the extraction of discriminative features from source subjects, and testify to the correctness of our re-implementation, a five-fold cross-validation subject-specific experiment is implemented. For each subject in the dataset and for each fold, 70% of the labeled data is used as the training data to construct a subject-specific classifier, while the remaining 30% of labeled data is used as a test dataset. As shown in Table 1, all baseline methods achieve competitive performance. This result is obtained without considering the possibility of baseline methods to underfit on source subject, further indicating the reason behind their suboptimal performance on the target dataset, namely, the inconsistency between source and target data distribution.

The CDAN model achieves the second best performance of 54.79% in accuracy, which is competitive among all baseline methods. CDAN implements adversarial domain adaptation to produce subject-invariant features, leading to relatively higher decoding performance. This result demonstrates the effectiveness of domain adaptation and the necessity of implementing feature transfer in cross-subject EEG decoding tasks. However, as can be seen from Table 2, subject 4 achieves an accuracy of 44.82% despite the use of domain adaptation. Comparatively, eSSN achieves 56.89%. By explicitly dividing the feature space into shared and private subspaces, eSSN is capable of constructing a shared feature space free of correlated noise, as demonstrated in [35]. Inspired by [35], the SSN’s architecture is consistent with the nature of cross-subject EEG decoding; therefore, achieving better results compared with the CDAN method.

## 5. Discussion

### 5.1. Visualization

To visualize the inconsistent characteristics of different subjects’ EEG features, all subjects’ mu wave-predictors are computed, and mapped to low-dimensional space to provide an intuitive perspective. A mu wave predictor is first proposed in [44] to evaluate subjects’ ability to use Motor Imagery BCIs. It is assumed that the subjects with high mu wave amplitude can activate more premotor areas and auxiliary motor areas related to motor imagination. Ref. [44] uses noise term and peak term to model the PSD of subjects, and the formula is expressed as follows:(9)$$g\left(f;\lambda ,\mu ,\sigma ,k\right)=g_{1}\left(f;\lambda ,k\right)+g_{2}\left(f;\mu ,\sigma ,k\right)$$
(10)$$g_{1}\left(f;\lambda ,k\right)=k_{1}+\frac{k_{2}}{f^{\lambda }}$$
(11)$$g_{2}\left(f;\mu ,\sigma ,k\right)=k_{3}N\left(f;\mu_{1},\sigma_{1}\right)+k_{4}N\left(f;\mu_{2},\sigma_{2}\right)$$
where *N* refers to Gaussian distribution and f stands for the frequency we consider in this experiment, which ranges from 4 to 40 Hz. λ is a parameter that controls the estimated noise curve. $g_{2}$ is the combination of two guassian distributions, representing the estimated mu activity. *N* refers to Gaussian distribution, and mu and theta are parameters of the distribution. This model is controlled by eight parameters, namely, $k_{1},K_{2},k_{3},k_{4},\mu_{1},\mu_{2},\sigma_{1},\sigma_{2}$, which can be estimated by using gradient descent. For each channel’s PSD data, the difference between the first peak value of PSD and the estimated function $g_{1}(f_{peak})$ at that peak value’s frequency is considered to represent the subject’s ERD amplitude, and used as that channel’s mu wave feature. After concatenating all 22 channels’ mu wave features, the final representation is mapped to two dimensions. As shown in Figure 5a, all nine subjects’ mu wave features are visualized in a two-dimensional plane, and each subject’s mu wave features form into a distinct cluster. It is significant that two of the subjects (Subject 8 and Subject 9) possess distinct mu wave features compared with the other subjects. This is consistent with the results shown in Figure 4c that the SSN trained on subject 8 achieved a better result on subject 9 compared with other subjects. It can also be seen that subject 2 and subject 7 shared similar distributions, while subject 1 and 6’s representations exhibit clear boundaries. Choosing one source subject only from the database in a random manner may lead to negative transfer, as in the case of subject 6 and 1. We solve this problem by introducing ensemble learning into the proposed framework. All source subjects’ information can be integrated into the final decision, thereby enlarging the learning capacity of the network.

For the purpose of presentation, target subject 1’s model trained on source subject 5 is adopted to produce the results shown in Figure 4b–f. We apply t-distributed stochastic neighbor embedding (t-SNE) to visualize the feature distribution before and after domain adaptation. t-SNE is a dimension reduction technique that reserves distance information between samples after transformation. For visualization, the feature distribution of baseline model and SSN are separated by a distinct offset. As can be seen from Figure 5b–f, baseline methods, including CSSP, FBCSP, EEGNet and ShallowConvNet, learn from each domain and exhibit relatively different feature distributions that have similar radius but a distinct center. On the contrary, with the implementation of domain adaptation, CDAN and eSSN share the same characteristics of the production of domain-invariant features, as source and target domain features converge into one cluster without an obvious boundary. This result demonstrates the effectiveness of domain adaptation to learn similar representations for source and target subjects.

### 5.2. Simulated Calibration Analysis

Although long-period calibration can lead to higher accuracy in subsequent sessions and higher ITR, it also results in a decrease of information delivered in total, as explained in Section 3.4. This paper aims to design a classifier that can achieve compatibly high classification performance without label information from the target domain, so as to reduce the calibration time. To demonstrate that our model requires less calibration data, we conduct a simulated calibration on each target subject’s test dataset. Of the trials, 20, 40, 60, 80% are used, respectively, to fine-tune the model sequentially, and the accuracy of the remaining parts are evaluated. The result in Figure 6a indicates that classification performance tends to rise as more trials are used for training. As shown in Figure 6b, eSSN achieves high performance in accuracy compared to CDAN and FBCSP. This is comprehensible, since eSSN is initially more accurate than other models for most of the subjects.

In Section 3.4, we demonstrate the concept of Session-ITR as a metric to evaluate cross-subject performance. Theoretical analysis gives an upper boundary for Session-ITR, which is further confirmed by our experiment. As seen from Figure 6c, the Session ITR of eSSN reaches its highest value when 60% of the trials are used as training data, then starts to drop when more data comes in. Meanwhile the single-trial accuracy of eSSN continues to grow. This phenomenon implicates that the loss of information caused by long calibration begins to outweigh the benefit brought by including more training data, which means the calibration phase should stop. The results of the simulated calibration experiment proved the usefulness of Session-ITR to objectively select a training ratio for specific application.

In the ablation study, we first considered using the deep convolution structures to extract high-order representations. We also tried to use a scale-invariant mean squared error term in the original version of [35]. All these structures proved to be suboptimal, as shown in Table 3. DeepConvNet achieves mean accuracy of 26.49%, while scale-invariant mean squared error fails to converge. This may be caused by the distinct nature between image data and EEG trials. For an image, the two dimensions of height and weight represent identical spatial features. While in EEG processing, the two dimensions contain different levels of features in separate domains. The operation to make scale invariant in the image domain may be invalid in EEG processing.

Many works on domain adaptation use Maximum Mean Discrepancy(MMD) loss to measure the difference between the source and target domain. We adopt ablation to study the MMD loss as a similarity loss that measures the distance between two subjects’ feature in the shared feature space:(12)$$L_{similarity}^{MMD}=\frac{1}{{\left(N^{s}\right)}^{2}}\Sigma_{i,j=0}^{N^{s}}\kappa \left(h_{ci}^{s},h_{cj}^{s}\right)-\frac{2}{N^{s}N^{t}}\Sigma_{i,j=0}^{N^{s},N^{t}}\kappa \left(h_{ci}^{s},h_{cj}^{t}\right)+\frac{1}{{\left(N^{t}\right)}^{2}}\Sigma_{i,j=0}^{N^{t}}\kappa \left(h_{ci}^{t},h_{cj}^{t}\right)$$
where κ refers to a linear combination of RBF kernel functions: $\kappa (x_{i},x_{j})$$=\Sigma_{n}\eta_{n}exp\{-\frac{1}{2\sigma_{n}}\left|\right|x_{i}-x_{j}{\left|\right|}^{2}\}$. Ns,Nt is the number of samples belonging to source domain and target domain, respectively; hci represents the cith feature in the source domain, and hcj represents the cjth feature in the target domain. The objective of using RBF kernel is to match all moments of the two populations using the Taylor expansion of the Gaussian function according to [35]. We also implement the MMD loss to constrain both domains’ features, but the loss fails to converge to a desired level. This may be caused by the high dimension of neural networks, which is distinct from traditional machine learning methods.

## 6. Conclusions

In this paper, we propose an ensemble Subject Separation Network domain adaptation framework to reduce the calibration time of MI-based BCIs. Firstly, we train the Subject Separation Network(SSN) for each source domain. A shared encoder was constructed by using adversarial domain adaptation. Private feature spaces orthogonal to the shared counterpart are learned from each subject. We also use a shared decoder to validate that the model is actually learning from task-relevant information. Finally, an ensemble classifier was built by integration of the SSNs using the information extracted from each subject’s task-relevant characteristics. To quantify the efficacy of the framework, we analyze the accuracy-calibration cost trade-off in an MI-based BCI, and theoretically guaranteed a generalization bound on the target error. The experiments on the public dataset fully demonstrate the superiority of the proposed method. On the other hand, the visualization of features from intermediate layers show that the source and target effectively form into one cluster after domain adaptation. In the future, we plan to improve the ensemble learning framework by integrating subject-specific resting-state EEG features.

As mentioned above, BCI technology can serve as assistive equipment and help motor-impaired patients to gain a certain level of autonomy in their daily life [6]. In this regard, the proposed framework can help to reduce the time of the traditional long-term training procedure of subjects to master MI-based BCIs, thus enabling confidential communication with relatives and friends with less effort. Nonetheless, the proposed methods also face crucial challenges to reach these goals. The Neural Network architecture of this eSSN framework is not applicable to other paradigms such as SCP or P300, which are commonly used for BCI communication applications. The non-stationary EEG feature is beyond the scope of this framework; and thus may induce drift problems in the long-term use of assistive BCIs by paralyzed users. We plan to further extend our framework by using more compact architectures and introducing adaptive methods to solve these problems.

The present work has the limitation of lacking a closed-loop experiment to verify the effectiveness of the proposed method. In the future we will extend the research by performing online evaluation of different domain adaptation methods to compare their capabilities of reducing calibration time for MI-based BCIs. Further, this work faces the limitation of only including offline processing of one public dataset. In the future, besides including more datasets into the offline process, we plan to conduct online experiments to test our method in more practical scenarios. An EEG Source Imaging technique will also be adopted to reveal the neurophysiological factors behind subject variation, to provide an intuitive means of understanding the mechanism of the domain adaptation process.

## Figures and Tables

**Figure 1 brainsci-13-00221-f001:**
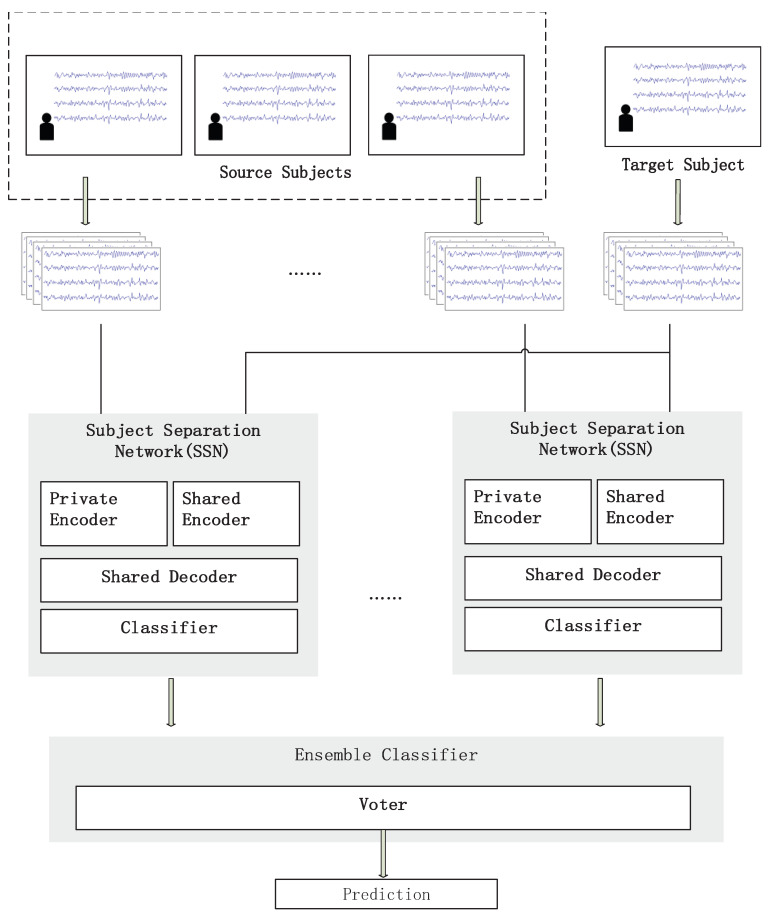
The proposed ensemble Subject Separation Network (eSSN) framework. The Subject Separation Network (SSN) is trained for each of the source–target subject pairs. All SSNs are then integrated to form a final decision, which is the predicted label of the target subject’s input trial.

**Figure 2 brainsci-13-00221-f002:**
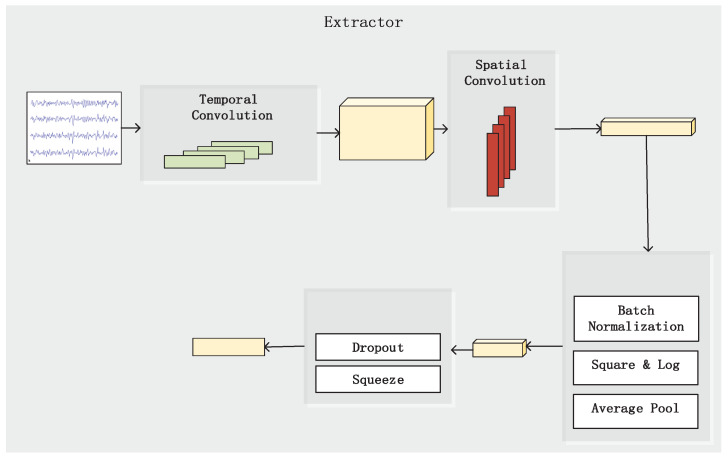
Detailed structure of the feature extractor. A 2D convolution is first applied to input signal to preprocess EEG signal through a set of time filters. Then a 2D convolution layer is concatenated behind to extract spatial-related information in the preprocessed signal. Finally after non-linear layer discriminative feature is extracted.

**Figure 3 brainsci-13-00221-f003:**
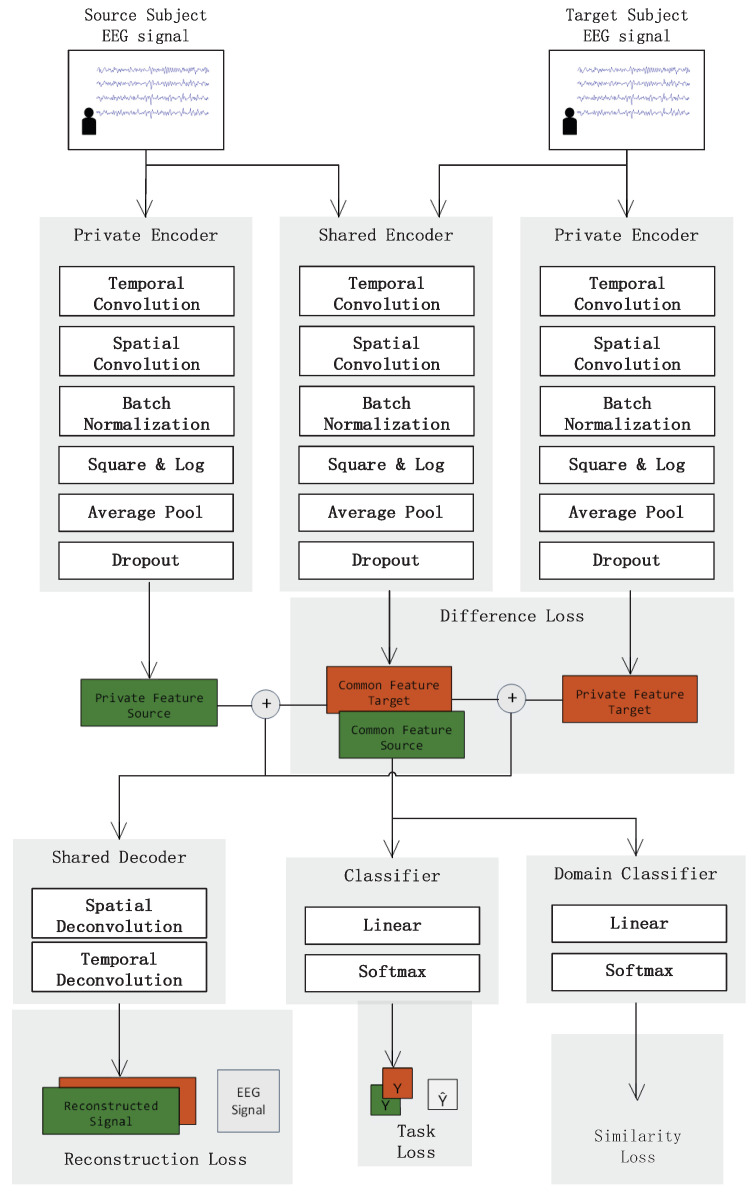
The overall architecture of Subject Separation Network (SSN). Four separate losses are constructed: Task Loss, Similarity Loss, Reconstruction Loss and Difference Loss. Task Loss is traditional cross-entropy loss. Difference Loss and Similarity Loss are added on features, while Reconstruction Loss is added on the original EEG trial and reconstructed signal. The Green color means feature or reconstructed signal comes from source space, and orange color means feature or reconstructed signal comes from target space.

**Figure 4 brainsci-13-00221-f004:**
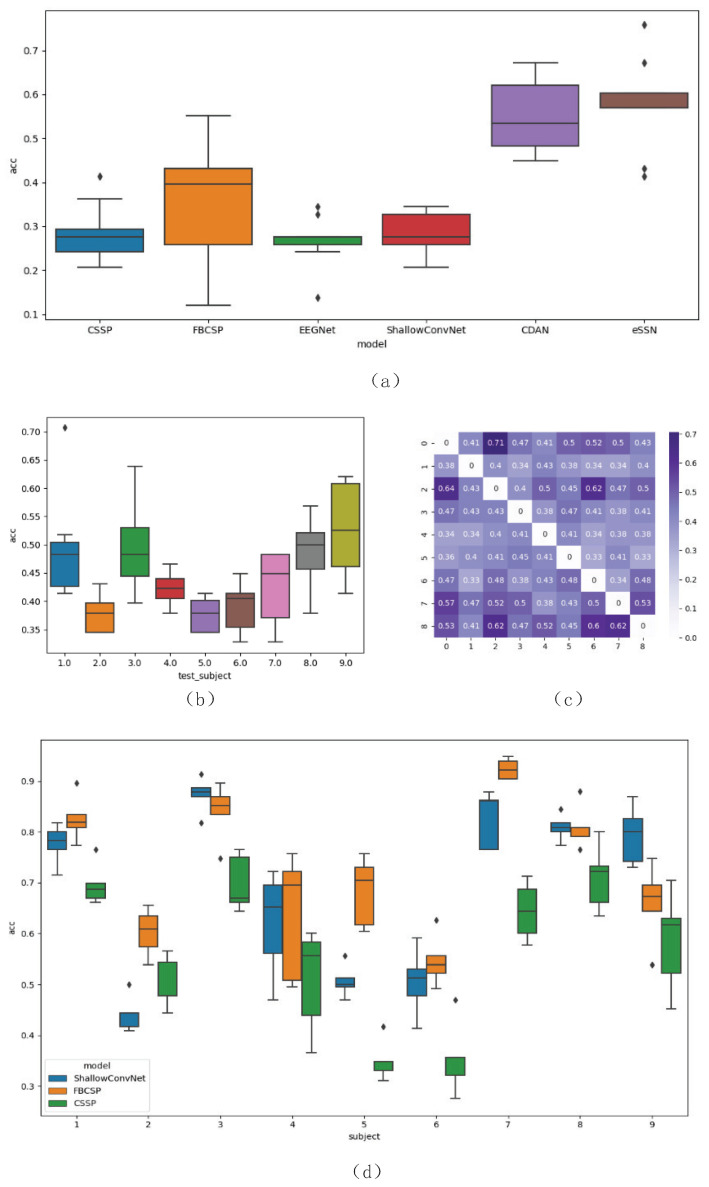
Results of the cross-subject and intra-subject experiments. The distribution of accuracy for the five-fold cross-validation is depicted as a box–whisker plot, the five lines in each box from top to bottom stand for the maximum value, the upper quartile, the median, the lower quartile and the minimum value, while the dots are outliers. The *y* axis in (**a**,**b**,**d**) stands for accuracy. (**a**) is mean accuracy of all models, averaged across all target subjects. (**b**) is the accuracy of each target subject’s SSNs. (**c**) is the heatmap of a matrix, representing the accuracy of each source–target subject pair’s SSN, while each row represents one target subject’s result. (**d**) is the result of five-fold cross validation within-subject experiment.

**Figure 5 brainsci-13-00221-f005:**
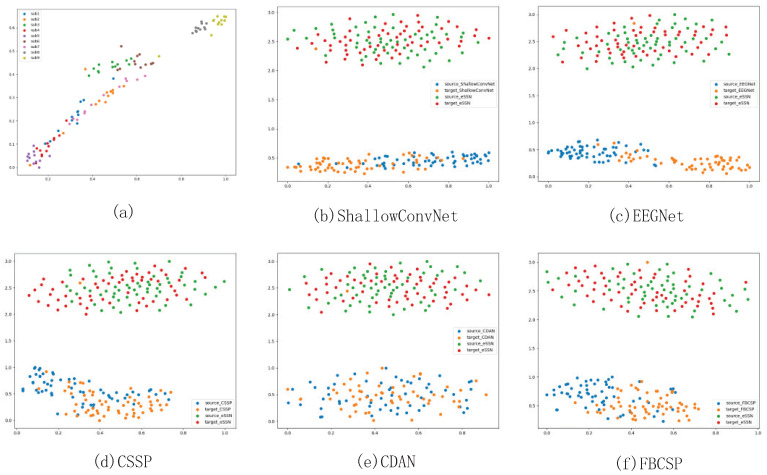
Visualization of comparison between the proposed method and the baseline models’ feature distribution. (**a**) is the visualization of each subject’s mu wave predictor, (**b**–**f**) is the visualization of each baseline model’s feature distribution compared to SSN. Target subject 1’s SSN trained on source subject 5 is used for demonstration.

**Figure 6 brainsci-13-00221-f006:**
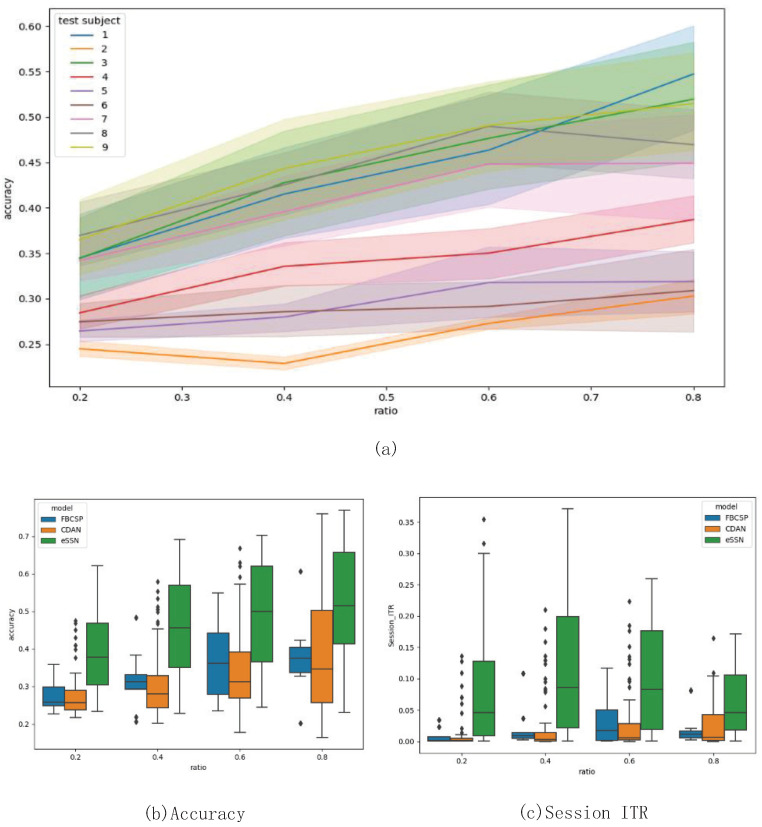
The performance of the proposed method in a simulated calibration experiment. In (**a**), standard deviations (SD) are depicted. The *x* axis stands for the ratio of trials used for training in the simulated calibration experiment. (**a**) is the distribution of the SSN’s accuracy over training ratio. (**b**) is the accuracy of FBCSP, CDAN and eSSN when diffrent amount of training data are adopted. (**c**) is the corresponding Session ITR.

**Table 1 brainsci-13-00221-t001:** Performance of baseline methods in intra-subject experiment. The values are the percentile mean accuracy and standard deviation of the five-fold cross-validation. “Shallow Conv” is abbreviation for Shallow ConvNet.

Subject	1	2	3	4	5	6	7	8	9
FBCSP	82.64 ± 4.47	60.23 ± 4.66	84.01 ± 5.64	63.56 ± 12.38	68.24 ± 6.84	77.61 ± 3.9	43.74 ± 3.73	87.32 ± 3.53	61.98 ± 10.4
ShallowConv	77.61 ± 3.9	43.74 ± 3.73	87.32 ± 3.53	61.98 ± 10.4	50.69 ± 3.19	50.53 ± 6.55	82.63 ± 5.62	80.89 ± 2.58	79.35 ± 5.82
CSSP	69.61 ± 4.12	50.16 ± 5.06	69.78 ± 5.56	50.88 ± 10.18	35.07 ± 4.03	35.6 ± 7.15	64.42 ± 5.69	71 ± 6.48	58.49 ± 9.82

**Table 2 brainsci-13-00221-t002:** Performance of the proposed method in cross-subject experiment. The values are the percentile accuracy on the test set. “Shallow Conv” is abbreviation for Shallow ConvNet. The largest value is marked with bold font.

Subject	1	2	3	4	5	6	7	8	9	Mean
CSSP	41.39	22.41	27.58	29.31	24.13	20.69	27.58	36.21	27.59	28.54
FBCSP	55.17	39.66	48.27	43.1	25.86	12.06	39.66	34.48	25.86	36.01
EEGNet	25.86	27.58	32.76	34.48	24.13	27.58	27.58	25.86	13.79	26.62
ShallowConv	25.86	27.58	32.75	34.48	24.14	20.69	27.58	32.76	29.31	28.35
CDAN	67.24	48.27	60.34	44.82	**44.82**	**50**	53.45	**62.07**	63.79	54.97
eSSN	**75.86**	**56.89**	**60.34**	**56.89**	41.37	43.1	**60.34**	56.89	**67.24**	**57.65**

**Table 3 brainsci-13-00221-t003:** Performance of DeepConvNet in cross-subject experiment. The values are the percentile accuracy. “Deep Conv” is abbreviation for Deep ConvNet.

Subject	1	2	3	4	5	6	7	8	9	Mean
DeepConv	32.65	25.34	31.56	35.21	22.01	18.56	24.26	30.24	18.59	26.49
proposed structure	75.86	56.89	60.34	56.89	41.37	43.1	60.34	56.9	56.9	67.24

## Data Availability

All datasets involved in this study is available online, from the MOABB database: https://github.com/NeuroTechX/moabb Codes of implementation can be found at: https://github.com/brandonlist/eSSN.

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
