# Peer review of "Subject Separation Network for Reducing Calibration Time of MI-Based BCI"

_brainsci, 2023, doi:10.3390/brainsci13020221_

Round 1

Reviewer 1 Report

This is a nice paper presenting a very interesting technique to aid in the inter-subject variation, which is intrinsic to any EEG processing pipeline.  The approach present a set of deep learning architectures to deal with this problem from the perspective of a solution of a transfer learning problem.  Authors tested their approach offline in a popular Motor Imagery dataset and the results show that their method achieves higher performance than other solutions.

The overall manuscript is ok.  There are just a minor problems to be reviewed/verified.

51: The concept of BCI-Illiteracy is quite discussed and complex.  I believe it is interesting to frame it a the complete lack of response from the BCI paradigm in terms of the signal response, and not some subject-specific adjustment on the signal.  It is something that cannot be solved by any signal processing method, because is the lack of the neurophysiological response.

72:"A shared decoder is tried to reconstruct the input...."  the meaning of this sentence is not clear.

72-77: This paragraph is really hard to understand, and it is not clear what you are proposing from it.

95: I insist with this concept.  Being illiterate, means being complete unable to perform the literate task.  Any intermediate form of system adaptation to subject-variabilities should not be considered bci-illiteracy. 

124: what do you mean by AROSAL ?

130: Please introduce SMR, sensorimothor rythms and their other name as mu wave.

135: there seems to be a problem with the writing in this sentence (beta band mentioned twice)

156: Be sure to define all the acronyms.

Figure 1: What is the meaning of the prediction by the end of the network.  Please clarify that.

217: Is it a 2D convolution ?   x is time points, times channels, times eeg amplitude.   3D ?

235: You should present both methods and later on in Results offer evidence why MMD loss is not compatible enough with EEG feature spaces.

338: The description of the datasets should be part of the method section, not results.

386: Is there any public repository where the implementation can be checked, in the name of reproducibility ? Ok found at the end, perfect.

Conclusion Section: please Authors should state more clearly the limitations of this work, in terms that it only includes offline processing of just one dataset.

Reviewer 2 Report

In this manuscript a subject separation network for reducing calibration time of MI-based BCI is proposed and the effectiveness of the proposed framework compared with multiple classification methods has been tested. In general, this is an interesting study which has the potential to improve current MI-based BCI systems. However, the following shortcomings must be addressed:

1.   Abstract: Please include in the abstract 1-2 sentences discussing your findings.

2.   Introduction and discussion sections: BCIs can not only be used for motor control (exoskeleton robot or wheelchairs) but also to regain autonomy in the interaction with the outside world e.g. by using a BCI based Web browser and enabling confidential communication with relatives and friends without an intermediator (a caregiver or a nurse). Please mentions these clinically crucial applications of BCIs and discuss the possible applications and limitations of your proposed framework to reach these goals. You can consider the following publications on this issue:    

A)   Karim, A. A. et al. (2006). Neural internet: Web surfing with brain potentials for the completely paralyzed. Neurorehabilitation and Neural Repair 20 (4), 508-515.

B)   Bensch, M. et al. (2007). Nessi: an EEG-controlled web browser for severely paralyzed patients. Computational Intelligence and Neuroscience. doi:10.1155/2007/71863

3.   Figures: In most figures the x- and y-axis are not readable. Please explain in the legends of figure 4 and 6 if standard deviations (SD) or standard errors of the mean (SEM) are depicted. All elements of the figures should be improved concerning their readability and understanding.

4.   Tables: Please explain in the legends of the tables the numbers (including their units) and the used abbreviations.

5.   Equations: Please check that all elements of the used equations are explained (e.g. equation 11 and 12).

 Author Response

Round 2

Reviewer 2 Report

The authors have adequately addressed all issues. Please only insert the references in the text.

Author Response

Dear Reviewer:

       The references mentioned in last round of review are added as reference 5 and 6 at the end of the manuscript. They are generated automatically thus I can not mark them as "modified". I'm sorry for not mentioning it in my cover letter.                                                   

Best regards,

Haochen Hu